# Key populations and healthcare providers perceptions, preferences and acceptability of HIV, Hepatitis B and C multiplex self-testing: A qualitative study

Elena Marbán-Castro[1]*, Caroline Thomas[2], Danil Nikitin[3], Catherine Thomas[2,3,4], Alla Bessonova[5], Sergei Bessonov[6], Claudius Mone Iye[2], Aibek Bekbolotov[7], Maia Japaridze[1], Mikaela Watson[1], Elena Ivanova[1], Olga Denisiuk[1], Sonjelle Shilton[1]

1 FIND, Geneva, Switzerland, 2 Peduli Hati Bangsa, Jakarta, Indonesia, 3 Global Research Institute Foundation (GLORI), Bishkek, Kyrgyzstan, 4 Universitas Jakarta Internasional, Jakarta, Indonesia, 5 Attika Harm Reduction NGO, Bishkek, Kyrgyzstan, 6 Kyrgyz Network of Harm Reduction (KNHR), Bishkek, Kyrgyzstan, 7 Kyrgyz Republican Center for Control of Blood Born Viral Hepatitis and HIV, Bishkek, Kyrgyzstan

* marbancastro.elena@gmail.com

## Abstract

Human immunodeficiency virus (HIV), hepatitis B virus (HBV), and hepatitis C virus (HCV) are chronic viral infections, sharing similar transmission routes. This study investigates the perceptions, preferences and acceptability of people who inject drugs (PWID) and healthcare providers (HCP) on multiplex rapid diagnostic tests (RDT) intended for self-testing (ST) for HIV, HBV and HCV. This was a qualitative study conducted in Indonesia and Kyrgyzstan. This study employed focus group discussions (FGDs) and semi-structured interviews (SSIs) to obtain general perceptions and theoretical acceptability on multiplex ST and cognitive interviews to optimize instructions for use (IFU). The IFU used were from four different manufacturers: PMC dual (HIV/HCV) ST in Kyrgyzstan, Mylab triple (HIV/HCV/HBV) ST in Kyrgyzstan, BioLytical (HIV/HCV), dual ST in Indonesia and Bioera Triple (HIV/HCV/HBV) ST in Indonesia. Eleven SSIs and twelve FGDs were conducted across both countries. A total of 110 participants were included in SSI and FGD: five stakeholders, 27 PWID, and 24 HCP in Kyrgyzstan and six stakeholders, 24 PWID, and 24 HCP in Indonesia. Participants reported the main factors that would facilitate the use of multiplex ST 1) quick time to results, 2) easy to use, 3) affordability, 4) confidentiality, and 5) ability to diagnose several infections at once. A total of 118 participants were included in cognitive interviews. Cognitive interviews led to recommendations for improving IFU, including larger fonts and simplified language. This study demonstrates initial acceptability of multiplex ST for HIV/HCV/HBV among PWID, HCP, and stakeholders. Still, communication

**Data availability statement:** Our protocol, as approved by the Ethics Committees of Bioethics under the Global Research Institute in the Kyrgyz Republic, on 04 April 2024 (reference number GLORI-IRB-117042024-1) and the Research Ethics Committee of Atma Jaya Catholic University of Indonesia, on 13 May 2024, (reference number 0004X/III/PPPE. PM.10.05/05/2024), did not include provisions for the public sharing of the full database generated during the study; neither did the participants' informed consent. As participants did not agree to us sharing their data publicly and to ensure compliance with national ethical guidelines of the countries and participant consent agreements, we did not include the dataset in a public repository. The data supporting the findings of this study are qualitative in nature and derived from in-depth personal interviews. Due to the potentially identifying content of full transcripts, sharing these data publicly would compromise participant anonymity and confidentiality, especially as our work included vulnerable groups such as people who inject drugs. Ethical restrictions on data sharing have been imposed by the committees that approved this study. In line with these restrictions, the full interview transcripts cannot be made openly available. Researchers who wish to request access to de-identified excerpts relevant to the study findings may contact us at the Foundation for innovative New Diagnostics (FIND) ("self-testing@finddx.org" or at "info@finddx.org", +4122 710 05 90, https://www.finddx.org/about-us/locations/geneva-switzerland/).

**Funding:** This study was funded by the National Institutes of Health (NIH) through the CRDF Global program, funded by the Government of the United States. The funders had no role in the study design, data collection, management, analysis, interpretation, manuscript writing, or the decision to submit the article for publication.

**Competing interests:** The authors have declared that no competing interests exist.

strategies, tailored IFU, targeted training to HCP and counseling to PWID are needed for future implementation.

## Introduction

Human immunodeficiency virus (HIV), hepatitis B virus (HBV), and hepatitis C virus (HCV) are the most common chronic viral infections globally [1]. They share common transmission patterns, as they are transmitted through blood, sexual intercourse, and mother-to-child [2].

HCV is the most common infection among people who inject drugs (PWID) [1]. PWID are a group exposed to increased risk of HIV and hepatitis acquisition and yet have limited diagnostic and treatment opportunities due to their stigmatized status. It is estimated that 7.4% of people living with HIV are co-infected with HBV and 6.2% with HCV, and the combined number of deaths from HBV, HCV and HIV exceeds 1.7 million annually [3].

Eastern Europe and Central Asia are regions with the highest HIV rates over the last decade [2]. Indonesia is one of the few countries with an increasing number of new HIV infections [4]. In Kyrgyzstan. HIV mainly affects populations at risk such as PWID [ 5]. Peer support interventions are key to enhance access to testing, linkage to and engagement in HIV care for PWID populations [6].

Self-care interventions, "the ability of individuals, families, and communities to promote health, prevent disease, maintain health, and to cope with illness and disability with or without the support of a health-care provider", are among the most promising new approaches contributing to universal health coverage [7]. Self-care has a high potential to complement and enhance traditional healthcare models. The ability to test at home reduces stigma associated with testing in the clinic, and bypasses issues with transportation and competing demands that may delay seeking care. A recent meta-analysis has shown that HIV self-testing is acceptable and may increase testing among those seeking emergency care, suggesting that HIV ST programs could enhance HIV diagnosis [8]. HIV self-testing has also shown to be acceptable and feasible among street adolescents in Togo [9].

In 2015, the WHO recommended HIV self-testing (ST) as an additional approach to increase the coverage of HIV testing services [10] and on HCV in 2021 [11]. Self-testing using multiplex immunochromatographic rapid diagnostic tests (RDTs) could be used as a complementary approach for HIV, HBV, and HCV, but its use has not yet been much explored [1]. There is scarce data on the acceptability of multiplex diagnostics for HIV and Hepatitis, by only two studies in the field [1,12]. There is one study on acceptability of a multiplex ST for HIV and syphilis, showing high acceptability from the 48 cisgender men and transgender women who have sex with men who participated in the study [13]. Overall, there may be multiple barriers that could prevent individuals from using these tools for screening, diagnosis, or for their use in different locations that have not been explored.

Rapid diagnostic tests (RDTs) can simultaneously detect HIV, HBV, and HCV, simplifying the testing process. Studies from sub-Saharan Africa have reported high

diagnostic accuracy and usability of triplex RDTs, though some misinterpretation of weak test bands has been observed [1]. Evidence from combined screening programs further suggests that offering all three tests together can increase case detection while adding only modest additional cost compared to single-disease testing [3]. This highlights the importance of generating further evidence on the usability, acceptability, and implementation of multiplex RDTs designed for self-testing.

We conducted a qualitative study which aimed to generate evidence on multiplex self-tests instructions for use (IFU), as well as perceptions from potential end users that can impact implementation outcomes such as potential barriers and facilitators in the deployment of HIV and Hepatitis ST.

## Methods

### Study design

This was a multi-country qualitative study conducted in Indonesia and Kyrgyzstan. Country selection was driven by an assessment of healthcare system structure, burden of HIV and viral Hepatitis, resource constraints and access barriers, and policy and regulatory framework.

### Location

In Kyrgyzstan, the study was conducted in the capital, Bishkek, by the Global Research Institute (GLORI) Foundation in collaboration with the harm reduction NGO, Attika, at the Attika headquarters. The Attika NGO provides psychosocial services to PWID, sex workers, individuals who were formerly incarcerated, people living with HIV, their partners and relatives. In Indonesia, the study was primarily conducted in the capital Jakarta, by Peduli Hati Bangsa in collaboration with the local NGOs and a network of people who use drugs, but also in Makassar (South Sulawesi) with the local NGO Drug User Network (Persaudaraan Korban Napza Makassar), and in Pontianak (West Kalimatan) with Yayasan Syafiya Putri Wiandra.

### Methodologies

This study employed three different methodologies, per stages, for the different objectives (Fig 1). Stage 1 included focus group discussions (FGDs) and semi-structured interviews (SSIs) to obtain general perceptions and theoretical acceptability on multiplex ST. Stage 2 included cognitive interviews to optimize IFU for 4 different manufacturers [6]. Results were shared with manufacturers to improve the IFU. The specific IFU were sent electronically by manufacturers and printed at the study sites using the same dimensions as used by the manufacturers.

Stage 1 : Qualitative data collection (FGD and SSI) to explore perceptions on multiplex ST:

2-3 FGD with 15-25 PWIDs (total).
2-3 FGD with 15-25 HCPs (total).
SSI with 3-6 stakeholders.

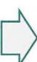

Stage 2: Cognitive interviews to optimize dual and triple multiplex ST instructions for use:

Dual ST with 15 PWIDs.
Dual ST with 15 HCPs.
Triple ST with 15 PWIDs.
Triple ST with 15 HCPs.

**Fig 1. Study design with types of populations for each country.** FGD: Focus Group Discussion; HCP: Healthcare providers; PWID: People who inject drugs; SSI: Semi-Structured Interviews; ST: Self-tests.

## Study population

Participants were eligible for inclusion in the study if they met all of the following inclusion criteria: aged 18 years or older, capacity to consent, speak the same language(s) as the facilitator, able to read (literate and sighted) and fulfil one of the following characteristics: 1) lived or living experiences of using injection drugs, 2) HCP, including social workers and others working at an harm reduction sites or other relevant local institution/NGO or 3) national stakeholder in public health measures, infectious disease control, surveillance, health systems etc., including but not limited to policymakers, medical experts and technology experts. Capacity was assessed by trained facilitators at the time of recruitment through verbal explanation of the consent form and confirmation of participant understanding. Participants who appeared unable to meaningfully engage in the consent process or the data collection at that time, for example due to acute intoxication or cognitive impairment, were not enrolled or were invited to participate at a later time if appropriate.

## Sampling and recruitment

Convenience sampling and snowball sampling were used via existing harm reduction sites and NGOs and methadone substitution therapy sites. Contact was made in person, either via email or phone. Stakeholders were contacted via focal contact points. The sample size calculations for the cognitive interviews were based on previous experience and a literature review. Cognitive interviews are usually conducted with small convenience samples [14]. For this study, a sample size of 10–15 participants per manufacturer's IFU was sought or until saturation was reached, whichever came first [15].

## Multiplex STs used

The IFU from the manufacturers were printed on the sites. Manufacturer-packaged IFU were not available at the time of data collection; instead, printed versions of manufacturer-provided electronic IFU were used. IFU from four different manufacturers were assessed: PMC dual (HIV/HCV) ST in Kyrgyzstan, Mylab triple (HIV/HCV/HBV) ST in Kyrgyzstan, BioLytical (HIV/HCV), dual ST in Indonesia and Bioera Triple (HIV/HCV/HBV) ST in Indonesia.

## Data collection

In-house piloting of questionnaires and the guides for the SSI and FGD was conducted prior to carrying out data collection. Sociodemographic data were collected via structured questionnaires. In Kyrgyzstan, data collection was conducted by an experienced research team (2 men, 1 woman, experienced researchers in HIV and Hepatitis, with previous training on qualitative research). Role-play sessions were conducted before data collection to refine interview guides and logistics. FGDs and SSIs were conducted in private rooms, cafes, or offices, with sessions averaging 67 and 58 minutes, respectively. Data were collected in Russian, audio-recorded, and securely stored. In Indonesia, data collection was conducted in Bahasa Indonesia by three researchers (1 men, 2 woman, experienced researchers in HIV and Hepatitis, with previous training on qualitative research). FGDs were held in cafes near harm reduction sites, while SSIs were conducted via Zoom. FGDs lasted an average of 85 minutes, and SSIs 52 minutes. Researchers and participants were alone for data collection with no other individuals present.

Cognitive interviews assessed participants' understanding of the IFU, with one-on-one sessions evaluating clarity and readability. Participants were also asked to score the IFU, being 0 the lowest score and 10 the highest and to discuss the chosen score. There was no prior relationship between researchers and participants; participants were explained the purpose of the research at the moment of the informed consent. The IFU rating was an exploratory comparative metric, not based on a validated scale. The output of the study was a structured set of user-identified comprehension and usability issues sent to the manufacturers. However, IFU were not revised by manufacturers and sent back to us as a result of this study.

To minimise social-desirability bias associated with peer-led data collection, interviewers explicitly emphasised the independence of the research from service provision, avoided recruiting participants with whom they had ongoing service relationships, and used neutrally framed questions in private interview settings.

## Data analysis

In Kyrgyzstan, qualitative data were transcribed semi-verbatim by local staff (A.B. and S.B., one woman and one man, experienced researchers in HIV and Hepatitis, with previous training on qualitative research), and verified by D.N. (men, experienced researcher in drug prevention, social work, HIV and Hepatitis, with previous training on qualitative research) to ensure completeness and accuracy. Data were analyzed thematically using Microsoft Excel. Researchers augmented field notes, identified key themes and deviant cases, and practiced reflexivity through regular meetings to minimize observer bias. Summaries were finalized in Russian and translated into English. In Indonesia, transcripts were fully verbatim done by local staffs (F.A., I.S., B.S., A.P., R.F., F.I., 3 women and 3 men), translated and anonymized by C.T (woman, an experienced HIV and hepatitis researcher with prior training in qualitative research, and coded thematically using Excel (C.T. and S.K.B, women, academics, experienced in qualitative research). For cognitive interviews, recordings supported field notes.

Rapid qualitative thematic and content analysis was conducted. Thematic analysis was conducted using a hybrid deductive–inductive approach, beginning with a preliminary codebook derived from the study objectives and interview guides. Codes were iteratively refined through double-coding of an initial subset of transcripts and regular analytic discussions. Themes were developed through comparison across participants groups and settings and finalized once thematic saturation was reached. Findings were shared with manufacturers to improve the ST IFU. Transcripts were not shared with participants, nor did they provided feedback on the findings. The data was reported in accordance with the CORE-Q guidelines [16] (Supporting Information S1 Fig) and the PLOS Inclusivity in Global Health research questionnaire (Supporting Information S2 Fig).

## Timeline

In Kyrgyzstan, data collection took place from the 4th of June until the 20th of June 2024. In Indonesia, data collection took place from the 5 of June until the 12 of August 2024.

## Ethical considerations

This study was conducted in accordance with the protocol, all International Council for Harmonization (ICH) and national regulations governing the conduct of clinical studies, ethical principles that have their origin in the Declaration of Helsinki, and all applicable local laws and regulations. Written informed consent was obtained before data collection. Data was analyzed anonymously. Ethical approval in Kyrgyzstan was obtained from the Committee on Bioethics under the Global Research Institute in the Kyrgyz Republic, on 04 April 2024, reference number GLORI-IRB-117042024–1. Ethical approval in Indonesia was obtained from the Research Ethics Committee of Atma Jaya Catholic University of Indonesia, on 13 May 2024, reference number 0004X/III/PPPE.PM.10.05/05/2024. There were no deviations from the study protocol after approval.

## Results

### Sociodemographic data of participants included in FGD and SSI (Stage 1)

In Kyrgyzstan, five SSIs were conducted with stakeholders, three FGDs were conducted with PWIDs, including 9 participants each, and three FGDs were conducted with HCP, including 8 participants in each. In Indonesia, six SSIs were conducted with stakeholders, three FGDs were conducted with PWIDs, including 8 participants each,

and three FGDs were conducted with HCP, including 8 participants in each. There were a total of 11 SSIs and 12 FGDs conducted. A total of 110 participants were included in the first stage of the study (Tables 1 and 2). In Kyrgyzstan, 56 participants were enrolled, including 5 stakeholders, 27 PWID and 24 HCP. Most participants were men: 60% of stakeholders (3/5), 59% of PWID (16/27) and 54% of HCP (12/24). Stakeholders were aged between 43 and 51 years, PWID between 19 and 40 years, and HCP between 32 and 45 years. In Indonesia, 54 participants were enrolled, including 6 stakeholders, 24 PWID and 24 HCP. Most participants were women in the SSIs (4/6, 67%) and in FGDs with HCP (13/24, 54%). However, most respondents in the FGDs with PWID were men (21/24, 87.5%). Stakeholders were aged between 33 and 67 years, PWID between 33 and 51 years and HCP between 29 and 55 years.

### Themes from participants included in FGD and SSI (Stage 1)

There were five main themes identified.

**Table 1. Sociodemographic characteristics of participants in FGD and SSI in Kyrgyzstan (Stage 1).**

| Type of participant | Stakeholders | | PWID | | HCP | |
|---|---|---|---|---|---|---|
| (data collection method) | (SSI) | | (FGD) | | (FGD) | |
| Number of participants | N = 5 | | N = 27 | | N = 24 | |
| Age (years) Mean [min, max] (SD) | 48 [43, 51] (3.0) | | 30.33 [19, 40] (5.9) | | 39.75 [32, 45] (4.0) | |
| | *n* | % | *n* | % | *n* | % |
| **Gender** | | | | | | |
| Men | 3 | 60 | 16 | 59 | 13 | 54 |
| Women | 2 | 40 | 11 | 41 | 11 | 46 |
| **Education** | | | | | | |
| Secondary | 0 | 0 | 6 | 22 | 2 | 8 |
| Vocational/college | 0 | 0 | 17 | 63 | 14 | 58 |
| University or higher | 5 | 100 | 4 | 15 | 8 | 34 |
| **Employment** | | | | | | |
| Full-time | 5 | 100 | 4 | 15 | 9 | 388 |
| Part-time | 0 | 0 | 12 | 45 | 10 | 42 |
| Homemaker | 0 | 0 | 2 | 7 | 0 | 0 |
| Self-employed/freelance | 0 | 0 | 7 | 26 | 3 | 12 |
| Unemployed | 0 | 0 | 0 | 0 | 2 | 8 |
| Student | 0 | 0 | 2 | 7 | 0 | 0 |
| **Residence** | | | | | | |
| Urban | 5 | 100 | 24 | 89 | 16 | 67 |
| Rural | 0 | 0 | 3 | 11 | 8 | 33 |
| **Ethnicity** | | | | | | |
| Kazakh | 1 | 20 | 4 | 15 | 5 | 21 |
| Russian | 1 | 20 | 8 | 30 | 5 | 21 |
| Korean | 0 | 0 | 1 | 4 | 1 | 4 |
| Kyrgyz | 3 | 60 | 9 | 34 | 9 | 37 |
| Tartar | 0 | 0 | 4 | 15 | 4 | 17 |
| Ukranian | 0 | 0 | 1 | 4 | 0 | 0 |

Table 2. Sociodemographic characteristics of participants in FGD and SSI in Indonesia (Stage 1).

| Type of participant | Stakeholders | | PWID | | HCP | |
|---|---|---|---|---|---|---|
| (data collection method) | (SSI) | | (FGD) | | (FGD) | |
| Number of participants | N = 6 | | N = 24 | | N = 24 | |
| Age (years) Mean [min, max] (SD) | 46 [33, 67] (13.1) | | 42 [33, 51] (3.9) | | 42 [29, 55] (7.0) | |
| | *n* | % | *n* | % | *n* | % |
| **Gender** | | | | | | |
| Men | 2 | 33 | 21 | 88 | 11 | 46 |
| Women | 4 | 67 | 3 | 12 | 13 | 54 |
| **Education** | | | | | | |
| Elementary | 0 | 0 | 2 | 8 | 0 | 0 |
| Secondary | 0 | 0 | 17 | 71 | 4 | 17 |
| Vocational/college | 1 | 17 | 3 | 13 | 9 | 37 |
| University or higher | 5 | 83 | 2 | 8 | 11 | 46 |
| **Employment** | | | | | | |
| Full-time | 6 | 100 | 6 | 25 | 22 | 92 |
| Part-time | 0 | 0 | 1 | 4 | 2 | 8 |
| Self-employed/freelance | 0 | 0 | 13 | 54 | 0 | 0 |
| Unemployed | 0 | 0 | 4 | 17 | 0 | 0 |
| **Residence** | | | | | | |
| Urban | 6 | 100 | 17 | 71 | 16 | 67 |
| Rural | 0 | 0 | 7 | 29 | 8 | 33 |
| **Ethnicity** | | | | | | |
| Batak | 1 | 17 | 2 | 8 | 1 | 4 |
| Betawi | 0 | 0 | 5 | 21 | 2 | 8 |
| Bugis | 2 | 33 | 4 | 17 | 8 | 33 |
| Java | 3 | 50 | 2 | 8 | 6 | 25 |
| Makassar | 0 | 0 | 3 | 13 | 0 | 0 |
| Malay | 0 | 0 | 5 | 21 | 3 | 13 |
| Others | 0 | 0 | 3 | 12 | 4 | 17 |

## Theme 1: Perceptions on self-testing and multiplex ST for HIV/HCV/HBV

Participants in both countries expressed that they have self-tested before for different conditions such as COVID-19, HIV, Hepatitis, influenza, cholesterol, blood sugar, pregnancy, and for alcohol and drug detection.

Issues with **confidentiality and potential discrimination**, as well as perceived inconvenient working hours, are factors that usually discourage people from getting tested.

*"Labs are open when I and my peers either work or sleep, for example, my preferred time is around 9pm but they don't operate at this time, and if I and my peers choose between sustaining work as a source of income and suggestion to test at a lab, testing won't be our priority."* (Kyrgyzstan, PWID, woman, 36 y.o.)

In Kyrgyzstan, HCP mentioned that groups categorized as "under-serviced", "hard to reach" or "under-tested" would benefit the most from access to multiplex ST. These groups included PWID, men who have sex with men, sex workers, people living with HIV, undocumented people and migrants. HCP and stakeholders in Kyrgyzstan stated that free professional use testing for HBV and HCV is widely available at government clinics and laboratories and positive individuals are

offered free treatment. Therefore, some HCP were skeptical about the use of these multiplex ST for the general population. Participants expressed that their **preference sample types** were saliva or urine because taking those samples were not invasive.

*"Not every health care professional has unique competencies of working with users, in my quite extended practice there were just two nurses who were able to properly locate my veins ruined by substances that I used to try when I was younger, therefore I am not quite sure that it is an easy thing to get even one drop of blood from a PWID."* *(Kyrgyzstan, PWID, man, 40 y.o.)*

Both HCP and PWID were open to consider blood and vaginal liquids for ST, if the first two options were not available.

*"It's simpler and faster, and does not require any significant preparations differently from blood and vaginal liquids, so if there is a choice, then saliva is the top preference, urine is number two, and then blood and liquids."* *(Kyrgyzstan, PWID FGD, woman, 29 years)*

In Indonesia, blood-based samples are the preferred type for self-testing across all participant groups, primarily due to perceptions of higher accuracy and comprehensive detection: *"I think blood is more accurate."* *(Indonesia, PWID FGD, man, 37 years)*

A tech specialist highlighted the efficiency of using one drop of blood for multiple tests, supporting the practicality of multiplex formats:

*"The idea of getting results for two or three tests with just one drop of blood is great."* *(Indonesia, HCP FGD, woman, 42 years)*

A stakeholder from Jakarta, Indonesia, highlighted that the introduction of comprehensive multiplex ST would mirror the success seen with the **dual test for HIV and syphilis**.

*"If there's a test for HIV and hepatitis, it will definitely have my full support because it mirrors the duo reagent [dual test] for HIV and syphilis, which has proven to be very helpful for the programme."* *(Indonesia, stakeholder, woman, 41 years)*

Participants in both countries understood that ST was **not intended to replace professional diagnosis,** but used as initial assessment, followed up by seeking professional medical advice.

*"It should be clear that this is a quick screening tool but does not replace a diagnosis from a medical professional or doctor."* *(Indonesia, stakeholder, man, 33 years)*

### Theme 2: Facilitators for multiplex ST for HIV/HCV/HBV

The main factors that could lead to people using multiplex ST according to participants in both countries were 1) short time in providing results, 2) easy to use, 3) affordability, 4) privacy and confidentiality to reduce stigma, 5) convenience and flexibility to test when needed and 6) ability to diagnose several infections at once.

**Sub-theme 2.1: Short time in providing results.** Participants appreciated the time efficiency and immediate results provided by ST. Moreover, stakeholders and HCP underscored the role ST could play in enhancing the efficiency of healthcare programmes, particularly in terms of **early detection and timely intervention.** HCP across various locations,

emphasized the convenience and flexibility that ST offers. They highlighted how ST allows individuals to perform tests at their own convenience, without the constraints of healthcare facility hours. The ability to test anytime and anywhere was particularly valued.

*"Yes, because you can do it anywhere, at any time, without being restricted by the hours of healthcare facilities."* *(Indonesia, HCP, man, 52 years)*

**Sub-theme 2.2: Easy to use.** HCP stated that as these ST are easy to use, those could be sold at several locations, reducing barriers in access.

*"It can be purchased anywhere, so if it is easy to use for testing, perhaps it could be sold at Indomarket." (Indonesia, HCP, man, 31 years)*

**Sub-theme 2.3: Affordability.** PWID participants stated that they would use multiplex ST if those were **"affordable"**. Among the PWID group in Jakarta, there was a clear demand for multiplex testing, especially if it could be made **affordable and accessible**. This reflected a pragmatic concern about the affordability of health interventions, which is crucial for their widespread adoption among marginalized communities such as PWID.

*"I hope that these three test tools can be available quickly in Indonesia and the price can be affordable and if possible subsidized by the government." (Indonesia, PWID, man, 41 years)*

**Sub-theme 2.4: Privacy and confidentiality to reduce stigma.** Benefits associated with multiplex ST included the **opportunity to avoid issues with confidentiality and potential discrimination**. The acknowledgement of the considerable level of **stigma** surrounding HIV testing further underscores the importance of providing a private, self-administered option for those who might otherwise avoid testing altogether. Participants from the PWID groups in both countries indicated that a key **motivation for ST is the desire to avoid the stigma** associated with visiting healthcare services. The fear of exposure and judgement drove the preference for ST, which they viewed as a means to keep their health status confidential. One participant noted that ST could avoid being recognized by people they knew at health facilities.

*"The accuracy is the same as a test at a health service like that, but sometimes they want to go to the health service and there tends to be concern because there are lots of people who know me." (Indonesia, PWID, man, 33 years)*

In Indonesia, participants described ST as a practical and private means of health monitoring, with an emphasis on **"confidentiality"**.

*"Self-test is a test that we do ourselves at home, right? Well, the advantage is that we can test at home and we can test ourselves, we and our confidentiality are guaranteed." (Indonesia, PWID, man, 37 years)*

The ability to test discreetly in their own spaces, such as in rented rooms or hotspots, further underscored the importance of privacy and convenience for this group.

*"Most prefer to use it in their rented rooms or at a hotspot where they hang out." (Indonesia, HCP, woman, 48 years)*

**Sub-theme 2.5: Convenience and flexibility.** The **flexibility** in testing locations and times, was considered to make ST an attractive option for diverse populations. HCP across various locations, emphasized the **convenience and**

**flexibility** that ST offers. They highlighted how ST allows individuals to perform tests at their own convenience, without the constraints of healthcare facility hours. The ability to test anytime and anywhere was particularly valued.

> *"Because you can do it anywhere, at any time, without being restricted by the hours of healthcare facilities." (Indonesia, HCP, man, 52 years)*

PWID participants also valued the ease of access to ST, especially if these tests were available in familiar, non-clinical settings, such as community centres, or through outreach programmes.

In Kyrgyzstan, participants suggested that if a dual ST was to be commercialized, they would prefer HBV and HCV combinations, rather than HIV and one viral hepatitis because HIV diagnosis, including HIV ST, is already working well.

> *"PWID are much less concerned about having HIV than hepatitis, especially PWID with prison background, that is why self-testing for HBV and HCV is more preferable." (Kyrgyzstan, HCP, woman, 42 years).*

**Sub-theme 2.6: Ability to diagnose several infections at once.** HCP, PWID and stakeholders in both countries expressed their preference for **multiplex ST containing a combination of three tests** (HIV/HCV/HBV) or many other infections as possible.

> *"Why test for 3 infections separately if it's very probable that they all will be detected, so let's test once." (Kyrgyzstan, HCP, woman, 35 years)*

Stakeholders appreciated the logistical benefits of ST, particularly multiplex ST, which allow simultaneous testing for multiple conditions, thereby simplifying the testing process and reducing the need for multiple tests. Many participants recommended that the design of multiplex ST should include sexually transmitted infections (STIs), such as chlamydia, gonorrhoea, trichomonas, and syphilis. However, they were also thinking strategically and acknowledged that it may be challenging to manage confirmatory testing and follow-up counselling and treatment for STIs at STI clinics.

> *"We all know that self-test results are preliminary results, and confirmatory testing is a compulsory and inevitable part of the whole diagnostic scope. So, it looks like it's not a problem in Kyrgyzstan to get support with HIV and hepatitis rather than with chlamydia and gonorrhoea as the venereological centres are rare and confirmatory testing in their labs will be quite costly." (Kyrgyzstan, HCP, woman, 45 years)*

Moreover, participants underscored the role ST could play in enhancing the **efficiency of healthcare programmes**, particularly in terms of early detection and timely intervention.

## Theme 3: Barriers for multiplex ST for HIV/HCV/HBV

The main factors that could pose challenges for multiplex STs use according to participants in both countries were 1) risk of results misinterpretation, 2) associated stigma, 3) lack of proper pre- and post-test support and counselling, 4) ST sensitivity, and 5) financial constraints and accessibility issues.

**Sub-theme 3.1: Risk of results misinterpretation.** In terms of concerns across all groups, a common concern was the **risk of inaccurate results, whether due to technical limitations, improper use, poor-quality test kits or lack of medical education to understand results**. Some participants in Kyrgyzstan had concerns about being able to **correctly interpret** multiplex ST results due to IFU that they found to not be user-friendly. They recommended translating the **IFU into local languages.**

*"Among our women clients in the south there are many who would be able to read only Uzbek and Tajik as they migrated from the neighbouring Uzbekistan and Tajikistan, they use drugs and are engaged in sex trading. Russian is universal of course, but developing materials in their native language will be of course helpful." (Kyrgyzstan, HCP, woman, 38 years)*

In Indonesia, participants reported a high demand for self-testing, but low trust in "unsupervised" use among PWID. PWID participants valued self-testing highly for privacy, confidentiality, and convenience, yet many simultaneously express fear of misuse, misinterpretation, and inaccurate results if tests are done without professional support.

*"In my opinion, self-testing is not good for drug addicts. Because I'm sorry drug addicts sometimes we have our own thoughts. They like to "experiment" with their own "medical science." [...] The point is, it would be better with professional people."* (Indonesia, PWID, male, 40 years)

**Sub-theme 3.2: Stigma.** The fear of **stigma and the potential psychological impact of receiving a reactive result without proper support** were common concerns shared in all groups across countries. Stakeholders expressed worry about the potential for misuse or unnecessary stress caused by ST without proper counselling, which could lead to severe consequences such as depression or suicide. Concerns about **stigma** may lead individuals to avoid purchasing ST at pharmacies, preferring the privacy of consulting a Puskesmas (primary healthcare center). Access to these services is time-sensitive, however, and flexible options such as online ordering or drop-in centres were suggested to improve accessibility. Additionally, HCP in Indonesia expressed concerns about the **psychosocial burden associated with purchasing ST kits from pharmacies,** where individuals might fear being judged. In contrast, accessing services through Puskesmas was considered to offer a more anonymous and supportive environment, which could encourage more people to make use of ST.

*"If they go to a pharmacy, there's definitely a psychosocial burden. They might worry about being judged because the staff will know they're buying a test for HIV or hepatitis." (Indonesia, HCP, man, 52 years)*

HCP highlighted that while ST may be easy to obtain, ensuring **anonymity** for accessing ST is crucial. They pointed out that many people might be hesitant to provide personal information, such as their national ID number or contact details, preferring to maintain their privacy when accessing these tests.

*"Not everyone is ready to be asked for the national ID or be assisted by a peer. Or even giving their contact number to someone they don't know. Some people like their privacy to be kept." (Indonesia, HCP, man, 35 years)*

The availability of **anonymous testing options** was seen as a facilitator, as it would allow individuals to test themselves without fear of exposure or **stigma**. Participants from the PWID group in Jakarta emphasized the importance of privacy in ST. They expressed concerns about the potential disclosure of individuals' HIV status, which could deter these individuals from using ST. The fear of stigma (including self-stigma) remained a major barrier. Stakeholders echoed these concerns, noting that privacy was a major consideration. They pointed out that while multiplex ST could help maintain privacy by allowing individuals to test at home, a **lack of appropriate education** and support could undermine the effectiveness of ST programmes.

Hepatitis C was perceived as an "easier" disease than HIV. PWID participants repeatedly differentiated Hepatitis C vs HIV. Hepatitis C self-testing is more acceptable, because: (1) Less stigma, (2) Clear follow-up pathway (SGOT/SGPT, referral hospitals), (3) HIV self-testing raises fear, especially related to disclosure, stigma, moral/religious judgement.

*"Yes, there is still too much stigma about HIV from the general public." (Indonesia, PWID, men, 41 years)*

*"Moreover with hepatitis… There's no stigma." (Indonesia, PWID, 46 years)*

**Sub-theme 3.3: Lack of support for pre- and pos-ST counselling.** Moreover, stakeholders stressed the need for **integrated counselling and referral services to support individuals who receive a reactive result**. They highlighted the importance of providing comprehensive information and online counselling options to guide users through the next steps to ensure they do not start treatment based on inaccurate or incomplete information.

*"Even though the results should be confirmed at healthcare facilities, the fear is that someone might start medication based on a self-test result without proper confirmation." (Indonesia, stakeholder, woman, 41 years)*

Some participants expressed **fears and uncertainties** about multiplex ST and the potential negative impact that it could have on some individuals with regards to their mental health. The solution suggested by PWID and HCP respondents was to develop and implement training courses for the staff of community-based NGOs who would be engaged in any multiplex ST implementation programmes.

*"For some reason, they [donors] take for granted that NGOs have enough capacity and are able to provide pre-post counselling that can reduce risk of depression – however, it is not the case at all, and this aspect has to be given more attention." (Kyrgyzstan, HCP, woman, 42 years)*

Stakeholders in Makassar and Pontianak, Indonesia, highlighted that while ST can be conducted at home, consulting a Puskesmas following a reactive result is vital to ensure proper **follow-up and care.**

*"For follow-up, individuals who test positive should go to the nearest Puskesmas. The Puskesmas will determine whether they can start treatment there or need to be referred to a specialized facility." (Indonesia, stakeholder, woman, 42 years)*

HCP in Indonesia raised concerns about the lack of follow-up support for individuals who received a reactive ST result. They emphasized the importance of having clear guidelines on where to seek further testing and treatment. The absence or lack of suitable follow-up could lead to missed opportunities for timely intervention and care.

*"There should be a clear follow-up process linked to healthcare services, and people should know where to go if they get a positive result." (Indonesia, HCP, man, 39 years)*

For the PWID group, **the lack of immediate follow-up** following an ST was seen as a disadvantage. Participants expressed concerns about the difficulty of navigating the healthcare system without proper guidance, especially when referrals or additional tests such as SGOT/SGPT (to assess liver function) were required.

*"Yes, after that, you have to do SGOT SGPT test. So, if you come without an escort and don't know the referral process, it's difficult." (Indonesia, PWID, man, 41 years)*

**Sub-theme 3.4: Sensitivity.** The accuracy of multiplex ST was also questioned, with concerns that their **sensitivity** may not be sufficient, leading to false-positive or false-negative results. The stakeholders further emphasized the need for clear standards, guidelines and public education to ensure that ST are used correctly and effectively. Stakeholders also expressed concerns that without proper education and support, ST programs may be ineffective, as many people in Indonesia only seek healthcare when they are already sick.

*"People often only seek healthcare when they're already sick... Without proper education and support, I don't think this self-testing program will be effective." (Indonesia, stakeholder, woman, 44 years)*

Participants from the PWID group in Jakarta, Indonesia, expressed concerns about the **availability and authenticity of ST.** One participant mentioned difficulties in accessing HIV ST, while another raised concerns about the proliferation of **counterfeit products in Indonesia, emphasizing the need to purchase ST from reputable sources** such as drug stores.

**Sub-theme 3.5: Financial constraints.** In both countries, participants mentioned that while multiplex ST could simplify logistics and make testing more accessible, a high **cost** in purchasing ST, or indirect derived costs, could be the main barrier of its implementation.

> *"According to the most recent internal assessment that our NGO conducted, there are around 35% of our regular clients who won't be able to pay anything for multiplex ST as they prefer to benefit from the lab testing even despite the inconveniences associated with it." (Kyrgyzstan, HCP, man, 45 years)*

> *"Even though the tests are free, there are still other costs like transportation and additional expenses beyond the diagnosis and treatment." (Indonesia, stakeholder, woman, 67 years)*

PWID participants in Makassar, Indonesia, highlighted **financial constraints** as a barrier, with some preferring to spend money on drugs rather than on test kits.

> *"For drug users, they'd rather spend their money on getting high than buying a test kit." (Indonesia, PWID, man, 46 years)*

### Theme 4: Service delivery for multiplex ST

In Kyrgyzstan, the preferred service delivery model to obtain multiplex ST was from: peer-based community-based NGOs, *"friendly"* pharmacies (open 24 hours a day 7 days a week) and men who have sex with men agencies. Whereas in Indonesia, preferred service delivery models included primary healthcare, pharmacies, convenient stores and online stores.

In Kyrgyzstan, participants reported that pre-test counselling should be provided, either in person or online, and that it could be given when multiplex ST were being distributed. According to the respondents, pre-test counselling at community-based NGOs must also include elements focusing on the prevention of incorrect interpretation of the IFU and the results by *"younger users"* with *"insufficient"* multiplex ST experience. Regarding post-test counselling, participants stated they would appreciate the opportunity to talk to an expert, including via teleconference after performing a multiplex ST at their home.

> *"I totally agree [on making self-test kits available at NGOs]. NGOs can help communities that need testing and provide information on treatment." (Kyrgyzstan, technology specialist, woman, 42 years)*

> *"Users who don't have experience, especially younger ones who even do not know the names of the diseases and infections, may commit certain mistakes and apply wrong manipulations. Therefore, having a trained nurse at NGOs who would demonstrate each step and manage pre-post in the clients' languages, is really important." (Kyrgyzstan, HCP, woman, 41 years)*

It was recommended to consider engaging **well-known individuals** in informational campaigns, such as celebrities, actors or politicians.

> *"Our athletes are doing pretty well at the Olympics, people love them, trust them and proud of them, so why not engage them [in a multiplex ST informational campaign] when they are back. But of course, they do not have to be supposed to issue multiplex ST kits to everyone passing by." (Kyrgyzstan, HCP, woman, 35 years)*

Regarding **communications channels** to receive ST information, respondents expressed their preference to use instant messaging apps, such as Telegram, as they considered them to be more accessible and less formal compared with websites. Telegram was also mentioned as a platform for the **procurement of multiplex ST.**

*"WhatsApp is worth of consideration, but Telegram (TG) is better… you know, TG-channels can be even used for procurement of multiplex ST as people can easily put an order and confirm payment details and delivery method." (Kyrgyzstan, PWID, man, 40 years)*

In Indonesia, stakeholders recognized the potential for Puskesmas (primary healthcare) to serve as **distribution points** for ST kits, particularly if this was integrated into government programmes.

*"If it's part of a government programme, they could place the kits in NGOs or at Puskesmas…" (Indonesia, stakeholder, man, 58 years)*

HCP in Indonesia shared a clear preference for **distribution through pharmacies**, as they are trusted sources for medical devices. However, HCP also considered the possibility of selling multiplex ST in more public and convenient locations, such as **local convenience stores** (e.g., Indomaret and Alfamart), suggesting that making the tests available in everyday retail settings could increase accessibility and encourage more individuals to test themselves. In Jakarta, there was a strong preference for **purchasing multiplex ST from pharmacies** due to concerns about the accuracy and reliability of tests bought online. These participants expressed skepticism about online purchases, fearing that the products might not be as accurate as tests purchased from pharmacies or could even be counterfeit. Conversely, participants from Makassar and Pontianak expressed a preference for **online availability** due to the anonymity it provides, which is crucial for individuals who might be uncomfortable purchasing multiplex ST in person. Some participants from Pontianak also stated they would be comfortable purchasing from **public places such as pharmacies**, recognizing them to be accessible yet sufficiently private to maintain confidentiality. Stakeholders in Jakarta were open to the idea of online sales, recognizing that online platforms allow users to check reviews and verify expiration dates, which are critical for ensuring reliability. A technology specialist from Jakarta noted that while ST are not yet widely available in supermarkets, it would be beneficial to expand their availability, potentially starting with online sales as users can verify product information.

When asked about mechanisms to support self-testing, PWID participants were unanimous: paper IFU alone are not sufficient. QR codes linking to videos, Instagram, or visual guides were strongly preferred.

*"Instagram. Apart from that, in my opinion there is a barcode so we just have to scan, if you scan it we can see how to use it." (Indonesia, PWID, Male, 41 years)*

**Theme 5: Acceptable costs for multiplex ST**

HCP reported that **co-payment** of the ST by potential buyers may increase linkage with service providers and clients' commitment to ST use. The maximum price that individuals would be willing to pay in Kyrgyzstan for multiplex ST mentioned by the HCP ($n=24$) and PWID respondents ($n=27$) varied from US\$2 to as much as US\$9 (average US\$5.65). It was suggested the cost of multiplex ST should be "maximally affordable" and comparable with, and no less than 25% lower, than laboratory-based testing.

*"If we go to test at the lab without referral sheet from the primary clinic, it will cost us 300 som [approximately $3.5], so if they [STs] are not around 25% cheaper than the lab price [i.e., available at $2.6], they may not be attractive to the end users. However, we're not sure whether this price will be attractive to the manufacturers." (Kyrgyzstan, PWID, man, 40 years)*

In Indonesia, participants from the PWID group generally suggested **prices** ranging from IDR 20,000 (US$ 1.2) to IDR 150,000 (US$ 9.6), depending on the type and number of diseases a test covered. A preference for lower prices was common, such as IDR 100,000 (US$ 6.4) for a triple test and IDR 75,000 (US$ 4.8) for a Dual test. There was also an emphasis on the value of accuracy, with some participants noting that IDR 100,000 (US$ 6.4) would be a fair price for a triple test if it provided reliable results.

*"If a test can detect multiple conditions and the price is slightly higher but still reasonable, it could be acceptable to people."* (Indonesia, stakeholder, woman, 37 years)

HCP in Indonesia expressed a strong preference for making multiplex ST as affordable as possible, ideally free of charge, to maximize accessibility. The notion of free distribution was further supported by a stakeholder in Makassar, who emphasized that all logistics and test distribution should come at no cost to the public, particularly for key populations and those at high risk. This perspective underscored the importance of affordability in ensuring that multiplex ST reach the populations who need them most.

### Optimization of IFU (Stage 2)

A total of 118 participants were included in Stage 2 (S1 Table). Results are presented by country and ST type in S1 to S4 Tables, to provide specific details for manufacturers regarding how to change and optimize their IFU. However, some common patterns were observed (Table 3). These common recommendations, as shown in the quotes below, included: 1) increase the font size to enhance readability, 2) shorten the IFU to include only essential information 3) provide more details about the control (C) line, and the implications of its absence, 4) avoid the use of technical terms, and 5) adjust the terminology of "positive" and "negative" results to avoid potentially misleading interpretations, as "positive" might be perceived to be a favorable outcome.

Table 3. Participants recommendations to strengthen the IFUs.

| Type of recommendation | Explanation | Illustrative quote |
|---|---|---|
| Formatting issues to be reviewed | • increased font size to enhance readability for text and images<br>• shorten IFU<br>• include a QR code or link with video instructions<br>• provide illustrations only for the materials contained within the kit (i.e., not scissors)<br>• better differentiation of pages (materials included in the kit, steps to conduct the test, results interpretation and actions after testing) | *"The text is too small for people with farsightedness. The layout is okay and the sequence or flow is clear."* (Indonesia, PWID, man, 47 years)<br>*"The layout and images are fairly clear […] It should be simplified, and page 2 should be made simpler."* (Indonesia, PWID, man, 43 years) |
| Content that requires further detail | • the control (C) line<br>• implications of the absence of the control (C) line<br>• implications of expiry date.<br>• actions in case bubbles appear while conducting the test<br>• actions in case more drops are added<br>• actions after results were provided (i.e., provide a local hotline or customer service number)<br>• safe disposal of the kit | *"It would make sense to add a stronger statement into the [Dual] IFU that if C line is missing, it is not only that the test is invalid - it also means that the results are invalid for each of the infections, and another test kit has to be used."* (Kyrgyzstan, HCP, woman, 40 years) |
| Terminology to be reviewed | • avoided technical terms such as "buffer", "POC"<br>• adjusted the terminology of "positive" and "negative" results | *"For example, put a larger figure on it or colour the bottle let say in green or yellow and refer to this item not as to a 'buffer vial' but let say 'green vial' or 'vial #1'."* (Kyrgyzstan, HCP, woman, 39 years)<br>*"In our culture 'positive' has a positive connotation like not infected. 'Negative' is something 'bad', therefore either the terms should be changed, or additional note has to be added explaining the background and meaning."* (Kyrgyzstan, HCP, woman, 45 years) |

Fig 2 represents participant's rating scores for the quality of the IFU per ST. In Indonesia, median (IQR) IFU scores were as follows: dual ST PWID 8.0 (7.5–9.0); dual ST HCP 8.0 (8.0–8.6); triple ST PWID 8.0 (7.0–8.0); triple ST HCP 8.0 (8.0–9.0). In Kyrgyzstan, median (IQR) IFU scores were as follows: dual ST PWID 6.0 (6.0–8.0); dual ST HCP 7.0 (6.0–8.5); triple ST PWID 6.0 (6.0–7.5); triple ST HCP 7.0 (6.0–7.0).

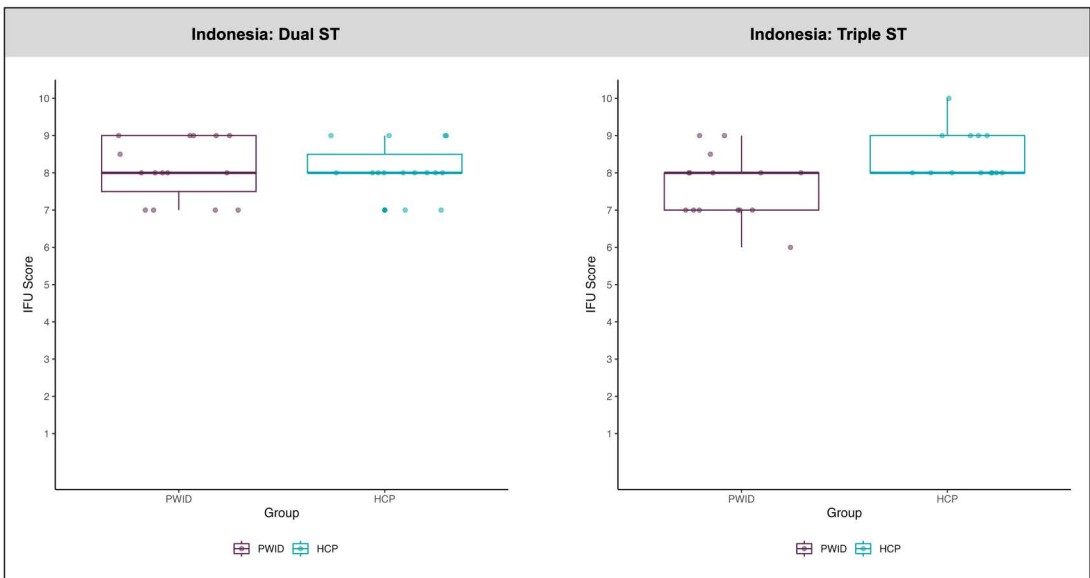

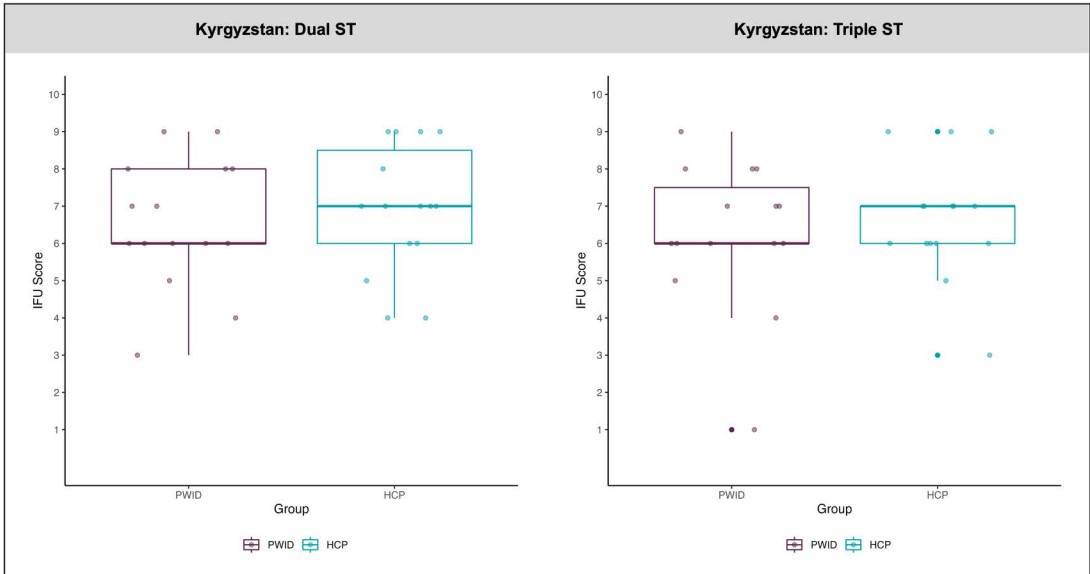

**Fig 2. Box plot of instruction for use (IFU) scores for the dual and triple ST in Indonesia and Kyrgyzstan between PWID and HCPs.** The plot displays the distribution of IFU scores, with the median represented by the central line in each box. The interquartile range (IQR) extends from the 25th to the 75th percentile, while the whiskers indicate the minimum and maximum scores within 1.5 times the IQR. Outliers are shown as solid (high opacity) points beyond the whiskers. All individual scores are plotted as lower opacity points to illustrate the data distribution.

## Discussion

This is the first study to explore end-user perspectives towards multiplex ST on HIV and viral h epatitis among key populations and HCP. This study included two stages with two different methodologies, which increased the richness of the data. Stage 1 study results indicate initial acceptability among PWID, HCP and stakeholders on multiplex ST for HIV, HBV and HCV. Participants expressed their preferences for multiplex ST combinations that include as many infections as possible, including HIV, HBV and HCV, and if possible STIs. Differences in service delivery models per countries were noted, with participants in Kyrgyzstan expressing their preferences for NGO and community-based organizations to reduce the stigma associated with accessing ST and being able to receive counselling, compared to participants in Indonesia where availability of ST in multiple locations, such as convenient stores, primary care, pharmacies and online, was preferred to reduce gaps in access. In both countries, peer-based pre- and post-test counselling was noted to be necessary to be provided. Stage 2 results indicate specific issues to be addressed in multiplex ST IFU to ensure they are comprehensible by potential users. Some of these suggestions could be relevant to other manufacturers, e.g., increasing the font size, providing a clearer explanation of the importance of the control (C) line and detailing specific actions to follow interpreting results. Additional materials, such as video instructions, could enhance user experience and uptake of ST. Recent studies have shown that displaying a video on HIV ST did not improve participant understanding of the IFU, but it increased participants confidence in the ability to self-test [17].There were some participants expressing concerns about the misinterpretation of results or lack of medical understanding by the general population. A study on HIV ST performance conducted among women sex workers in Uganda indicated concordance among participants self-interpretation and manufacturer instructions of 73% (95% CI 56% to 86%) for HIV-reactive self-tests and 68% (95% CI 54% to 80%) for HIV-not reactive self-tests [18]. New technologies that include interpretation of ST results might be key to solve knowledge gaps in interpretation of results [19,20] and to provide trust to end-users. Further research will be essential to contextualize the factors influencing the results from the triple ST IFU in Indonesia, and to extend the investigation to additional population groups and geographical areas. Regarding the potential cost of the ST, participants in both countries expressed that ST should be ideally free (subsidized), include copayment strategies or have a low cost. The ST cassettes used as part of this study were similar to the ST and RDTs already available in local medical facilities and pharmacies [21]. ST and RDTs for HIV and HCV have also recently been made available in medical facilities, NGOs and pharmacies [22].

A study conducted in Malawi and Zambia used cognitive interviews to explore interpretation of HIV ST IFU and concluded that difficulties with IFU were noted by participants with lower literacy levels and those from rural settings [23]. In Indonesia, a study on community-based HIV ST among man clients of sex work revealed that only 13% of them had ever tested for HIV, but their acceptability for HIV ST during the study was of 65% [24]. In a 2019 qualitative study involving 29 men clients of sex work in Bali, Indonesia, only 31% had previously tested for HIV [25]. Based on the evidence that participants preferred ST due to privacy issues and the commitment to reach 95% HIV case detection by 2025, the Indonesian government launched community-based HIV screening, including ST, as part of a broader public health strategy to target key and general populations [26]. In the Technical Guidelines, the main distribution channels for community-based HIV screening services were: 1) NGO offices or drop-in centers, designated hotspots, or agreed-upon locations with clients, 2) public and private healthcare facilities including pharmacies and trained community health workers, and 3) workplaces via in-house medical staff, safety officers, or trained peer counselors as part of workplace HIV programs.

Our study had some limitations. First, this study was conducted in two countries, which provides geographic representation of two very different contexts, with different backgrounds and experiences regarding ST. However, they are not representative of all LMICs or all the countries in their region. Second, at the time we were collecting data, the multiplex ST were not available in Kyrgyzstan or Indonesia. Thus, HIV/HCV/HBV multiplex ST are not currently commercially available in Kyrgyzstan or Indonesia; therefore, the research participants only discussed their theoretical acceptability, as well as the acceptability of the preliminary IFU issued as part of this research effort. The SSIs and FGDs were based on hypothetical questions and participants' previous experiences, instead of an actual scenario where participants were able to see

the multiplex ST and read the IFU. Limitations from the cognitive interviews include that the interviews were performed with printed versions of the IFU. Differences in participants' viewing of the real instructions might happen and might differ from study results. Also, as the interviews were not conducted with the ST, it might have been difficult for participants to interpret some steps, in the absence of a physical kit. Selection bias could also be present due to the convenience and snowball sampling methods used. Additionally, the settings in which the assessments were conducted may have influenced participants' comfort levels and responses. Given the close involvement of harm-reduction organisations and peer networks in recruitment, self-reported acceptability may be subject to residual social-desirability bias and should be interpreted as reflecting perceptions within established harm-reduction networks rather than all PWID. Observed differences in IFU ratings should be interpreted as indicative of relative participant preference within the study sample rather than as generalisable or externally comparable usability scores. This study was not designed to assess real-world usability, nor comprehension of IFU associated with ST execution, including finger-prick difficulty, blood volume adequacy, timing, and error recovery. All these limitations should be considered when interpreting the results.

Subsequent research should incorporate observed ST with full kits to assess procedural errors, invalid results, and concordance between perceived and actual usability under routine-use conditions. Further implementation pilots should also be conducted to assess the best service delivery models for these multiplex ST to reach those in need.

These results have been disseminated with stakeholders and experts in multiplex ST, policymakers and programme implementors in an in-person meeting organized by FIND and WHO in Geneva in November 2024.

## Conclusions

This study shows the initial acceptability of multiplex ST for HIV and viral hepatitis among PWID, HCP, and stakeholders. However, communication strategies are needed to increase acceptability, particularly as blood-based tests were the least preferred. Integration into existing services alongside pharmacy and community-based distribution could enhance access. Peer-based counseling and tailored IFU, including video instructions, may improve user experience and uptake. Further research is needed to understand barriers and facilitators before multiplex ST implementation.

## Supporting information

**S1 Table. Sociodemographic characteristics of participants in the optimization of IFU in Kyrgyzstan and Indonesia (Stage 2).**
(DOCX)

**S1 Fig. CORE-Q checklist.**
(PDF)

**S2 Fig. PLOS Inclusivity in Global Health research questionnaire.**
(DOCX)

## Acknowledgments

We would like to thank the study participants for their time and willingness to contribute to this research. We further thank the World Health Organization.

## Author contributions

**Conceptualization:** Elena Marbán-Castro, Olga Denisiuk, Sonjelle Shilton.

**Data curation:** Elena Marbán-Castro, Mikaela Watson.

**Formal analysis:** Elena Marbán-Castro, Caroline Thomas, Danil Nikitin, Catherine Thomas, Alla Bessonova, Sergei Bessonov, Claudius Mone Iye, Mikaela Watson.

**Funding acquisition:** Elena Ivanova, Olga Denisiuk, Sonjelle Shilton.

**Investigation:** Caroline Thomas, Danil Nikitin, Catherine Thomas, Alla Bessonova, Sergei Bessonov, Claudius Mone Iye, Aibek Bekbolotov, Maia Japaridze, Elena Ivanova, Olga Denisiuk, Sonjelle Shilton.

**Methodology:** Elena Marbán-Castro, Caroline Thomas, Danil Nikitin, Catherine Thomas, Alla Bessonova, Sergei Bessonov, Claudius Mone Iye.

**Project administration:** Caroline Thomas, Danil Nikitin, Catherine Thomas, Alla Bessonova, Sergei Bessonov, Claudius Mone Iye, Maia Japaridze.

**Resources:** Elena Marbán-Castro.

**Supervision:** Elena Marbán-Castro, Aibek Bekbolotov, Maia Japaridze, Elena Ivanova, Sonjelle Shilton.

**Validation:** Elena Marbán-Castro.

**Visualization:** Mikaela Watson.

**Writing – original draft:** Elena Marbán-Castro.

**Writing – review & editing:** Caroline Thomas, Danil Nikitin, Catherine Thomas, Alla Bessonova, Sergei Bessonov, Claudius Mone Iye, Aibek Bekbolotov, Maia Japaridze, Mikaela Watson, Elena Ivanova, Olga Denisiuk, Sonjelle Shilton.

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
