## [Decision Letter · Decision Letter 0]

12 Jan 2026

PONE-D-25-62011Key populations and healthcare providers perceptions, preferences and acceptability of HIV, Hepatitis B and C multiplex self-testing: A qualitative studyPLOS One

Dear Dr. Marbán-Castro,

Thank you for submitting your manuscript to PLOS ONE. After careful consideration, we feel that it has merit but does not fully meet PLOS ONE’s publication criteria as it currently stands. Therefore, we invite you to submit a revised version of the manuscript that addresses the points raised during the review process.

We look forward to receiving your revised manuscript.

Kind regards,

Yury E Khudyakov, PhD

Academic Editor

PLOS One

Journal Requirements:

5. We note that you have indicated that there are restrictions to data sharing for this study. PLOS only allows data to be available upon request if there are legal or ethical restrictions on sharing data publicly. For more information on unacceptable data access restrictions, please see http://journals.plos.org/plosone/s/data-availability#loc-unacceptable-data-access-restrictions.

Additional Editor Comments:

Your manuscript was reviewed by two experts in the field, who identified many important issues in your submission and produced detailed comments. Please review the attached comments and provide point-by-point responses.

Reviewers' comments:

Reviewer's Responses to Questions

Comments to the Author

1. Is the manuscript technically sound, and do the data support the conclusions?

Reviewer #1: Partly

Reviewer #2: Yes

2. Has the statistical analysis been performed appropriately and rigorously? 

Reviewer #1: N/A

Reviewer #2: Yes

3. Have the authors made all data underlying the findings in their manuscript fully available?

Reviewer #1: No

Reviewer #2: Yes

4. Is the manuscript presented in an intelligible fashion and written in standard English?

Reviewer #1: Yes

Reviewer #2: Yes

5. Review Comments to the Author

Reviewer #1: The manuscript is written in a scholarly manner with standard English used. However, there are some areas in the different sections of the manuscript that need clarity, as indicated in the attached comments, to make it read better. I have suggested some revisions to the Methods, the Results and Discussion.

Reviewer #2: This is a well-conducted and timely qualitative study that addresses an important gap in the acceptability of multiplex self-tests for HIV, HBV, and HCV among key populations in two high-burden settings. The multi-method approach, direct engagement of PWID, HCPs, and stakeholders, and practical feedback to manufacturers on IFUs are notable strengths. The manuscript is clearly written and provides valuable insights for implementation.

Please a summary of the strengths and few areas of improvement.

Strengths

• This is a Multi-country and Multi-method Design and Conducting the study in two diverse settings (Indonesia and Kyrgyzstan) provides comparative insights into cultural, healthcare, and policy differences, enhancing the applicability of findings to low- and middle-income countries (LMICs). The use of FGDs, SSIs, and cognitive interviews adds depth and triangulation, increasing data richness and reliability.

• The Inclusion of Key Populations and Stakeholders and Engaging PWID (a hard-to-reach group), HCP, and policymakers ensures voices from end-users and implementers are represented, making the results practical and relevant for real-world deployment. This is encouraging to see such a study prioritizing these marginalized groups

• Great and Practical Recommendations for Manufacturers: Cognitive interviews led to actionable IFU improvements (e.g., larger fonts, simplified language, clearer control line explanations), which were shared directly with manufacturers, demonstrating direct impact on product design. (Improvements suggested below )

• The Focus on Underserved Issues and Addressing a literature gap on multiplex ST acceptability for HIV/hepatitis, with emphasis on barriers like stigma and counseling needs is commendable, this aligns with WHO self-care guidelines and contributing to universal health coverage goals.

• The analysis on Comprehensive Facilitators and Barriers is good and Identifies nuanced preferences (e.g., blood vs. saliva samples by country) and service delivery models (e.g., NGO-based in Kyrgyzstan vs. pharmacies/online in Indonesia), providing a holistic view for policy and program design.

Recommendations for Improvement

• Enhance Clarity and Consistency of Sociodemographic Reporting Table 2 contains formatting errors and incomplete data (e.g., age means and ranges are misaligned across columns, percentages sometimes do not add up correctly). Clean and double-check the table for accuracy before final publication to avoid reader confusion.

• Provide More Detailed Description of Cognitive Interview Findings The abstract and discussion mention that cognitive interviews led to recommendations (larger fonts, simplified language, better explanation of control lines), but the manuscript (in the provided pages) does not yet include a dedicated results subsection or table summarising the specific problems identified for each of the four manufacturers’ IFUs. Adding a concise table or bullet-point list of the most common comprehension issues and the exact changes suggested per brand would greatly strengthen the paper and make the practical contribution more visible.

• Include the Revised IFUs or Visual Examples Since the study’s major tangible output was improved IFU that were returned to manufacturers, consider including (in supplementary material) before-and-after examples of the most problematic sections (e.g., result interpretation diagrams or text passages). This would allow readers to see the concrete impact of participant feedback.

• Strengthen Potential Bias Discussion The researchers in Kyrgyzstan were affiliated with the harm-reduction NGO partner (Attika), and in Indonesia worked closely with local drug-user networks. While this peer-led approach is a strength for access and trust, explicitly discuss how this proximity was managed to minimise social-desirability bias in responses (e.g., participants possibly overstating acceptability to please familiar organisations).

• Improve Transparency on Saturation The sample size for cognitive interviews was planned at 10–15 per IFU “or until saturation”. State clearly whether saturation was reached for each of the four IFU and, if not, note which aspects remained uncertain.

• Minor Methodological Clarifications

o Specify exactly how the IFU scores (0–10) given by participants were analysed and whether average scores per manufacturer are available.

o Briefly explain why no actual self-test kits were used (only printed IFUs) and acknowledge this as a limitation for assessing real-world usability (e.g., finger-prick difficulty, blood volume, timing).Or just state this clearly as a limitation and may be a recommendation for future studies.

With these minor revisions, the manuscript will be suitable for publication in PLOS ONE.

Thank you for the opportunity to review this work.

6. PLOS authors have the option to publish the peer review history of their article (what does this mean?). If published, this will include your full peer review and any attached files.

Do you want your identity to be public for this peer review? For information about this choice, including consent withdrawal, please see our Privacy Policy.

Reviewer #1: No

Reviewer #2:  Yes: Helgar Musyoki

---

## [Author Response · Author response to Decision Letter 1]

16 Feb 2026

FIND

Global Health Campus

Chemin du Pommier 40, 1218 Le Grand Saconnex

Geneva,

Switzerland

Dear Editor,

Thank you for your feedback on our manuscript entitled Key populations and healthcare providers perceptions, preferences and acceptability of HIV, Hepatitis B and C multiplex self-testing: A qualitative study (PONE-D-25-62011).

Please find our detailed responses to each of the comments received.

Comments from Academic Editor:

Authors’ response: Thank you for this suggestion. However, this is not applicable to our qualitative study.

Authors’ response: Thank you for this suggestion. We have reviewed style requirements and adapt it accordingly.

Authors’ response: We have now uploaded the questionnaire on inclusivity within Supporting information files.

Authors’ Response: We appreciate this important clarification. We have deleted all the funding information from the manuscript.

Authors’ Response: We have reviewed the funding information to reflect it accurately.

5. We note that you have indicated that there are restrictions to data sharing for this study. PLOS only allows data to be available upon request if there are legal or ethical restrictions on sharing data publicly. For more information on unacceptable data access restrictions, please see http://journals.plos.org/plosone/s/data-availability#loc-unacceptable-data-access-restrictions.

Authors’ Response: We appreciate this important clarification. We have deleted the funding information from the manuscript. Due to ethical restrictions, the absence of participant consent for public data sharing, and safety reasons (potential identification of participants) full qualitative transcripts cannot be made available. De-identified analytic materials, including the coding framework, codebook, and anonymized exemplar quotations, cannot be shared publicly, but can be made available upon reasonable request to the corresponding author and subject to institutional approvals.

Authors’ Response: We have included captions for supporting information files at the end of the manuscript.

Review Comments to the Author

Reviewer #1: The manuscript is written in a scholarly manner with standard English used. However, there are some areas in the different sections of the manuscript that need clarity, as indicated in the attached comments, to make it read better. I have suggested some revisions to the Methods, the Results and Discussion.

This is an interesting qualitative study on ‘Key populations and healthcare providers perceptions, preferences and acceptability of HIV, Hepatitis B and C multiplex self-testing’. It is demonstrating initial acceptability of multiplex self-testing of HIV, Hepatitis B and C among people who inject drugs, health care providers, and stakeholders, which is aimed at generating evidence on instructions for use, as well as perceptions from potential end users that can impact implementation outcomes.

Generally, the manuscript is well written but below are a few suggestions to help improve its robustness.

1) My observation is that there are quite many abbreviations. For a phrase/word that appears very few times in the manuscript, I suggest you do not abbreviate. Also, abbreviations need to be expounded at first use, and some abbreviations need to be expanded for the reader to grasp meaning e.g. HCW (Pg 11, line 2)

Authors’ Response: We completely agree with this reviewers’ comments. We have reviewed the manuscript and deleted some abbreviations that were not very much frequent in our manuscript. For others, such as “IFU”, “ST”, “HIV” etc. we will need to leave them to enhance readability. Thanks for realizing that we once mentioned “HCW”, for healthcare workers, it was a mistake and we have changed it to “HCP”, as we described it earlier “healthcare providers”.

2) Please indicate in your methods, what guided your sample selection? Was the 120 for cognitive interviews an additional sample or was this the same sample? This needs to be clearly reflected. How did the author(s) define ‘capacity to consent’?

Authors’ Response: We thank the reviewer for this important clarification. We have revised the Methods section to explicitly describe the sampling strategy and eligibility criteria. Specifically, we now state that purposive sampling was used to recruit PWID, healthcare providers, and stakeholders based on their relevance to the study objectives. We have also operationalized “capacity to consent” as the ability to understand the study information, communicate a voluntary decision, and meaningfully participate in data collection, as assessed by trained facilitators at the time of recruitment. This criteria was specifically relevant for the PWID population that we were working with. In case that facilitators perceived that participants were under the effects of drugs, they were not able to be informed about the study, nor participate in data collection. This clarification has been added to the Methods section to improve transparency and reproducibility.

For the cognitive interview component, the target sample size of 10–15 participants was defined a priori as a maximum, with thematic saturation assessed concurrently during data collection. Saturation was operationalised as the point at which no new issues related to comprehension, interpretation, or perceived usability of the IFU content emerged in successive interviews. Saturation was reached for all four IFUs. As a result, data collection did not exceed the upper bound of the planned sample size. We have revised the Methods section to clarify that the sampling rule was “whichever came first” (saturation or 10–15 interviews per IFU) and have added a brief statement in the Limitations section noting which aspects of IFU usability may remain uncertain despite achieving saturation on core domains.

3) How many FGDs were conducted and what was the size of the FGDs?

Authors’ Response: Thank you very much for realizing that this information was not implicit in our manuscript. In Kyrgyzstan, five SSIs were conducted with stakeholders, 3 FGDs were conducted with PWIDs, including 9 participants each, and 3 FGDs were conducted with HCP, including 8 participants in each. In Indonesia, six SSIs were conducted with stakeholders, 3 FGDs were conducted with PWIDs, including 8 participants each, and 3 FGDs were conducted with HCPs, including 8 participants in each. There were a total of 11 SSIs and 12 FGDs conducted. We have included this explicitly in the abstract and in the results section.

4) Pg 6 still in the methods, under ‘Instructions on use’, clearly explain what you mean by ‘No real IFUs were used’. Did participants have any experience of RDTs? If not, were participants asked to read the IFUs printed on the sites to ensure they read them? The authors should clearly indicate how this was done to ensure participants had information for the cognitive interviews.

Authors’ Response: Thank you very much for realizing this. At the time of data collection, manufacturer-packaged instructions for use (IFUs) were not available. Manufacturers provided electronic versions of IFUs, which were printed locally at each study site and used during the qualitative sessions. IFUs were printed with the same final formatting, paper quality and font size provided by the manufacturers. However, there might be slight differences with the printed IFUs by manufacturers provided in the self-test kits.

Participants’ prior experience with rapid diagnostic tests (RDTs) varied by respondent group and setting; prior RDT use was not a prerequisite for participation. These STs were not available on the market for their use.

During cognitive interviews, participants were asked to review the printed IFU in real time and to verbalize their understanding of each step, consistent with standard cognitive interviewing techniques aimed at identifying comprehension barriers, misinterpretations, and usability challenges. No additional verbal instructions beyond the printed IFU were provided, in order to assess comprehension based solely on the written materials.

We acknowledge that IFU comprehension may differ when using final manufacturer-packaged materials; this limitation is discussed explicitly in the Limitations section.

5) The last statement under the ‘Data collection section’ needs revision to clearly indicate which people were part of the activity since the ‘No one …’ is confusing

Authors’ Response: Thank you for this input. We have reviewed the text accordingly. This comment responds to CORE Q guidelines to provide information if other people were present during data collection. We have clarified that no one else was present apart from participants and researchers.

6) Regarding the Analysis; who transcribed/verified the data? The authors need to clearly show how the data verification process was done to ensure its credibility and completeness. I suggest that the authors chronologically indicate how they identified the keys themes/codes.

Authors’ Response: To ask partners to explain further. We thank the reviewer for this suggestion. We have revised the Analysis section to explicitly describe (i) who transcribed and verified the data, including transcript review against audio recordings to ensure completeness and accuracy, and (ii) the chronological analytical process used to develop codes and themes. The revised text now details the use of a hybrid deductive–inductive thematic analysis, initial codebook development, double-coding of a subset of transcripts, iterative refinement through team discussions, and theme finalization based on constant comparison and thematic saturation.

7) In the Results section, paragraph 2 under ‘Perceptions on self-testing…’, line 1 of this paragraph indicates ‘inconvenient working hours’…the authors need to elaborate more on this by indicating the time that participants considered inconvenient and why.

Authors’ Response: Thank you so much for this comment. We have included a quote in the manuscript to explain further what participants meant by “perceived inconvenient working hours”. Please, find the quote below: "Labs are open when I and my peers either work or sleep, for example, my preferred time is around 9pm but they don't operate at this time, and if I and my peers choose between sustaining work as a source of income and suggestion to test at a lab, testing won't be our priority". (Kyrgyzstan, PWID, woman, 36 y.o.)

8) Still under the Results, Table 1 looks congested and not easily readable. Secondly, it does not reflect the total number (overall) of participants to include participants in cognitive interviews. If these were the same participants for SSIs and FGDs, this needs to be clearly come out under ‘sampling and recruitment’ in the methods section.

Authors’ Response: Thanks to this comment we have reviewed the Table and created two separate tables for the sociodemographic data of participants in the interviews per country. We have also included the sociodemographic information from those participating in cognitive interviews in the Supporting information as Table S1.

9) I suggest that the authors systematically organise the results to clearly indicate the major themes and sub-themes and clearly organise the findings under the relevant themes.

Authors’ Response: Thank you for such an important point. We have restructured the Results to make them easier for the readers.

10) Generally, the PWID voices do not seem to be strongly coming out of the data. We need to have more of their voices on the various aspects in the results.

Authors’ Response: Thank you very much for realizing this important aspect. We have included additional quotes from PWID participants to best reflect their responses.

11) The results for stage 2 need to have more depth, with supportive quotes following detailed analytical text of the findings instead of having stand-alone quotes.

Authors’ Response: Thank you very much for realizing this. We have created a Table for better explanation. Additionally, we have reviewed the results and added additional findings in this section.

12) In your Discussion, it would be good for the comparison of results from the two countries to appear in the results section rather than in the discussion. The discussion here should focus on the study findings in relation to other studies/ existing literature.

Authors’ Response: We appreciate this important comment. We ha

---

## [Decision Letter · Decision Letter 1]

22 Mar 2026

Key populations and healthcare providers perceptions, preferences and acceptability of HIV, Hepatitis B and C multiplex self-testing: A qualitative study

PONE-D-25-62011R1

Dear Dr. Marbán-Castro,

We’re pleased to inform you that your manuscript has been judged scientifically suitable for publication and will be formally accepted for publication once it meets all outstanding technical requirements.

Kind regards,

Yury E Khudyakov, PhD

Academic Editor

PLOS One

Additional Editor Comments (optional):

Reviewers' comments:

Reviewer's Responses to Questions

Comments to the Author

1. If the authors have adequately addressed your comments raised in a previous round of review and you feel that this manuscript is now acceptable for publication, you may indicate that here to bypass the “Comments to the Author” section, enter your conflict of interest statement in the “Confidential to Editor” section, and submit your "Accept" recommendation.

Reviewer #1: All comments have been addressed

Reviewer #2: All comments have been addressed

2. Is the manuscript technically sound, and do the data support the conclusions?

Reviewer #1: Yes

Reviewer #2: Yes

3. Has the statistical analysis been performed appropriately and rigorously? 

Reviewer #1: N/A

Reviewer #2: Yes

4. Have the authors made all data underlying the findings in their manuscript fully available?

Reviewer #1: No

Reviewer #2: Yes

5. Is the manuscript presented in an intelligible fashion and written in standard English?

Reviewer #1: Yes

Reviewer #2: Yes

6. Review Comments to the Author

Reviewer #1: The manuscript flows smoothly and I wish to thank the authors for incorporating the comments. However, the authors need to thoroughly read through the manuscript and fix a few typo and grammatical errors and punctuation (e.g. pg 6, the first sentence in the last paragraph or pg 7 & 8, where you need to match the numbers with the noun’s plurality – 1 man, 2 women). Other than that, I do not have any more comments.

Reviewer #2: This manuscript presents a robust qualitative study on the acceptability of multiplex self-tests for HIV, HBV, and HCV among PWID, HCPs, and stakeholders in Indonesia and Kyrgyzstan. The multi-method approach, including FGDs, SSIs, and cognitive interviews, effectively captures perceptions and optimizes IFU, addressing a critical gap in self-testing implementation. Strengths include diverse participant engagement, practical manufacturer feedback, and alignment with WHO guidelines. Overall, the work offers valuable insights for public health strategies and merits publication in PLOS ONE

7. PLOS authors have the option to publish the peer review history of their article (what does this mean?). If published, this will include your full peer review and any attached files.

Do you want your identity to be public for this peer review? For information about this choice, including consent withdrawal, please see our Privacy Policy.

Reviewer #1: No

Reviewer #2: Yes: Helgar Musyoki

---

## [Editor Report · Acceptance letter]

PONE-D-25-62011R1

PLOS One

Dear Dr. Marbán-Castro,

I'm pleased to inform you that your manuscript has been deemed suitable for publication in PLOS One. Congratulations! Your manuscript is now being handed over to our production team.

Kind regards,

on behalf of

Dr. Yury E Khudyakov

Academic Editor

PLOS One